# Frontal Transcranial Direct Current Stimulation as a Potential Treatment of Parkinson’s Disease-Related Fatigue

**DOI:** 10.3390/brainsci11040467

**Published:** 2021-04-08

**Authors:** Tino Zaehle

**Affiliations:** 1Department of Neurology, Otto-von-Guericke-University Magdeburg, 39120 Magdeburg, Germany; tino.zaehle@ovgu.de; 2Center for Behavioral Brain Sciences (CBBS), 39106 Magdeburg, Germany

**Keywords:** fatigue, Parkinson’s disease (PD), tDCS

## Abstract

In contrast to motor symptoms, non-motor symptoms in Parkinson’s disease (PD) are often poorly recognized and inadequately treated. Fatigue is one of the most common non-motor symptoms in PD and affects a broad range of everyday activities, causes disability, and substantially reduces the quality of life. It occurs at every stage of PD, and once present, it often persists and worsens over time. PD patients attending the 2013 World Parkinson Congress voted fatigue as the leading symptom in need of further research. However, despite its clinical significance, little progress has been made in understanding the causes of Parkinson’s disease-related fatigue (PDRF) and developing effective treatment options, which argues strongly for a greater effort. Transcranial direct current stimulation (tDCS) is a technique to non-invasively modulate cortical excitability by delivering low electrical currents to the cerebral cortex. In the past, it has been consistently evidenced that tDCS has the ability to induce neuromodulatory changes in the motor, sensory, and cognitive domains. Importantly, recent data present tDCS over the frontal cortex as an effective therapeutic option to treat fatigue in patients suffering from multiple sclerosis (MS). The current opinion paper reviews recent data on PDRF and the application of tDCS for the treatment of fatigue in neuropsychiatric disorders to further develop an idea of using frontal anodal tDCS as a potential therapeutic strategy to alleviate one of the most common and severe non-motor symptoms of PD.

## 1. Introduction

Fatigue is a complex symptom and a multifaceted construct that leads to a general feeling of exhaustion, loss of motivation, and behavioral performance problems [1]. It is a major cause of traffic accidents [2] or accidents in other work-related settings [3]. Importantly, fatigue is often comorbid to a variety of neuropsychiatric disorders, such as depression, cancer, multiple sclerosis (MS), and Parkinson’s disease (PD).

In patients with PD, fatigue is one of the most common non-motor symptoms affecting a wide range of daily activities, leading to disability, and significantly reducing the quality of life [4]. Despite its clinical importance, progress in understanding and treating fatigue is still remarkably limited. Some therapeutic approaches for fatigue in PD have been tested, but none are effective against fatigue. Conventional therapies for the motor symptoms of PD do not significantly improve fatigue [5]. 

While transcranial direct current stimulation (tDCS) has recently been shown to alleviate fatigue in multiple sclerosis (MS) effectively [6,7,8], data on fatigue in PD are sparse. 

In the current opinion paper, I will propose that NIBS approaches can contribute to a better understanding of the fatigue syndrome and stimulate the development of efficient treatments based on rational hypotheses about the underlying pathophysiology, and, finally, argue for frontal anodal tDCS as a potential therapeutic option in Parkinson’s disease-related fatigue (PDRF).

## 2. Parkinson’s Disease-Related Fatigue (PDRF)

Parkinson’s disease (PD) is the second most common neurodegenerative disorder, affecting approximately 1% of the population over 50 years of age [9]. PD is traditionally defined as a basic motor disorder. However, many non-motor symptoms (NMS) also commonly occur in PD. These NMS include pain, cognitive decline, delusions, and notable fatigue. Among the NMS deficits, Parkinson’s disease-related fatigue (PDRF) in particular is one of the most common symptoms in PD. It affects up to 58% of patients [10,11], and 30% of PD patients report that PDRF is the symptom with the greatest negative impact on their daily lives [4]. Accordingly, PDRF is an important stressor with a tremendous negative impact on the patients’ quality of life and an essential contributor to disease burden [12,13,14]. Moreover, PDRF already impacts patients at an early untreated stage of the disease and is an important consideration in patient management [15].

In general, from the Latin *fatigare*, fatigue describes an overwhelming feeling of tiredness, weakness, lack of energy, and exhaustion unrelated to physical activity [16]. In various neurological diseases, fatigue is an important but often underappreciated complaint [17,18]. 

Nowadays, patients suffering from PD are usually appropriately treated for their motor symptoms, whereas a significant proportion of NMS still remains unrecognized or unreported [19]. However, despite the current diagnostic underrepresentation, NMS were described at the very beginning of the clinical description of the syndrome [20]. The first description of PDRF likely came from J. M. Charcot, who described fatigue as early as in the 1870s, in addition to several other typical non-motor aspects of PD [21]. Thus, although recognizing the importance of PDRF seems to be a relatively recent development, it was already recognized in the nineteenth century by the most important clinical neurologists of their time.

Although fatigue is a common and debilitating symptom in PD, the exact etiology and underlying pathophysiology of fatigue in PD remain unclear [22], and—accordingly—there is a significant lack of available effective treatments for PDRF [5]. This considerable lack of progress in understanding fatigue’s pathophysiology and its treatment is, in part, due to the fact that fatigue still lacks a universally accepted definition and classification [23,24].

This lack of a consistent fatigue taxonomy complicates its understanding, measurement, and consequently its treatment [25]. To date, fatigue is mostly assessed subjectively using self-report questionnaires. However, because patients assess their perceived fatigue symptoms retrospectively, self-assessments of fatigue are subject to regression to the mean and recall errors that may reduce their accuracy. For example, available fatigue questionnaires for disease-related fatigue in multiple sclerosis (MS) showed low correlations with each other and heterogeneous associations with patients’ functional impairments, disease duration, or cognitive deficits [26,27,28]. In contrast to these subjective fatigue measures, a fatigue-related decline in performance—also known as fatigability—could be quantified using objective indices [29]. Thus, to overcome the subjective nature of fatigue measures and the associated limitations for diagnosis and intervention of MS-related fatigue, we and others [25,30] proposed a generalized fatigue taxonomy that is disease nonspecific and universally applicable. Here, fatigue was broadly classified into physical, psychosocial, and cognitive fatigue. While psychosocial fatigue can only be assessed subjectively, physical and cognitive fatigue concepts imply that fatigue can be assessed both qualitatively as a subjective phenomenon and quantitatively as an objective phenomenon [25]. Specifically, subjective cognitive fatigue refers to a persistently perceived feeling of exhaustion. In contrast, objective cognitive fatigue—also referred to as fatigability—refers to a decline in performance on cognitive tasks, quantifiable as a change in cognitive performance relative to a baseline [23]. Finally, subjective and objective cognitive fatigue can be further subdivided. Subjective fatigue is divided into a trait component and a state component. Trait fatigue refers to a global status that changes slowly over time, whereas state fatigue refers to the change in subjectively perceived fatigue level over time [31]. Accordingly, subjective trait fatigue can be assessed by self-questionnaires and subjective state fatigue by visual analog scales (VAS) or numerical rating scales. In contrast, objective fatigue (fatigability) is, by definition, state-dependent and allows an objective assessment by behavioral or electrophysiological parameters. 

Analogous to the assessment in patients with MS, an objective fatigue diagnosis appears to be a prerequisite for information and education in the early disease management of patients with PD [15,32] and ultimately for effective treatment of PDRF.

### 2.1. Ethology of PDRF

The inconsistencies in fatigue definitions also negatively affected the understanding of the pathophysiology of PDRF [23]. Despite the enormous negative impact of fatigue in PD, it remains challenging to delineate the pathophysiology of PDRF from other NMS in PD. In general, proposed physiologic mechanisms include increased circulating proinflammatory cytokines, dysfunction in nigrostriatally and extrastriatally dopaminergic pathways, involvement of non-dopaminergic (especially serotonergic) pathways, autonomic nervous system involvement, and, importantly, underlying prefrontal pathology [33,34,35]. 

Previously, PDRF was often assumed to be a reactive phenomenon [36]. In fact, PDRF is highly related to the severity of depressive symptoms [15,37]. Therefore, the understanding of PDRF is significantly biased by its co-occurrence with affective disorders [38]. However, a recent comprehensive review of PDRF [39] summarized clinical and experimental findings that support the view that fatigue is a primary manifestation of PD and not a secondary phenomenon. Accordingly, although PDRF is consistently associated with depression in PD, depression and fatigue often exist independently, and fatigue may persist after a successful depression treatment [40]. In fact, PDRF is present in over 50% of non-depressed PD patients [10]. Moreover, PDRF may precede motor symptoms [41] and does not necessarily correlate with PD duration or motor disability [36]. Thus, PDRF does not appear to be systematically associated with disease duration, stage, or motor symptoms; does not correlate with objective motor fatigability; and is distinguishable from other affective symptoms such as depression, apathy, and somnolence. In addition, PDRF does not respond reliably to dopaminergic or surgical therapies [42,43,44]. This evidence suggests that PDRF is a primary symptom in PD and is related to pathological nonmotor networks [36].

Recent hypotheses on pathophysiological mechanisms suggested that specific dysfunctions in the frontal cortex may play a significant role in fatigue. Evidence for the involvement of frontal lobe dysfunctions came from observations of fatigue-related executive impairment in patients with PD [22,45]. Accordingly, PDRF was associated with decreased frontal lobe blood flow [22] and prefrontal hypoperfusion [46]. Additionally, impaired connectivity within the frontal lobe was associated with PDRF [47]. This observed hypoactivation of the frontal lobe fitted well with a general model of pathological fatigue [32] that assumed central fatigue as a consequence of dysfunction in a circuit involving the basal ganglia and the frontal cortex. Analogously, Clayton and colleagues [48] introduced an oscillatory model of sustained attention, in which frontomedial theta power supported cognitive control processes while alpha power over task-relevant cortical areas suppressed task-irrelevant processes. They postulated that when a person becomes fatigued, both frontomedial theta and alpha power over task-relevant areas increase. The increase in frontomedial theta power may reflect the reactive engagement of theta-driven cognitive control processes via low-frequency phase synchronization. In contrast, the increase in alpha power over task-relevant cortical areas (e.g., occipital in a visual attention task) suppressed information processing and caused attentional deficits. According to Clayton et al. (2015), the increase of frontomedial theta power reflected the detection of a mismatch between current and desired levels of attention and, in turn, acted as a compensatory control mechanism to enlarge top-down control processes in a fatiguing brain. 

### 2.2. Treatment of PDRF

According to the 2018 review of the International Parkinson and Movement Disorder Society (MDS) Evidence-Based Medicine (EBM) committee, which regularly publishes recommendations on treating Parkinson’s disease nonmotor symptoms, only the monoamine oxidase (MAO)-B inhibitor rasagiline was considered possibly useful for the management of PDRF when other secondary causes of fatigue were excluded [49]. The efficacy of methylphenidate and modafinil remained investigational. However, a recent comprehensive review on current pharmacologic and non-pharmacologic treatment options for PDRF came to a less positive evaluation. The authors concluded that there was insufficient evidence for all treatment strategies. However, among the available options, the best evidence appeared to be for doxepin, rasagiline, and levodopa infusion therapy [50]. Finally, some studies indicated supportive effects of deep brain stimulation of the subthalamic nucleus (STN-DBS) on PDRF. In an open multicenter study including 60 patients [51], as well as in a subsequent international multicenter, observational study on 173 PD [52], STN-DBS could significantly improve NMS, including fatigue. However, there were also contradicting reports showing that fatigue could also be commonly caused by DBS surgery in PD [43] or at least could not be excluded on an individual level [53]. 

## 3. Transcranial Direct Current Stimulation (tDCS)

As PDRF drastically affects the patients’ quality of life, the development of efficient therapeutic methods for fatigue treatment is of high clinical relevance. Furthermore, for a systematic treatment evaluation and optimization, a reliable and valid assessment of the individual fatigue level by objective parameters is essential. 

Transcranial direct current stimulation (tDCS) may offer a unique opportunity to manipulate the maladaptive neuronal activity underlying PD-associated fatigue. The neuromodulatory potential of tDCS was widely demonstrated for cognitive, perceptual, and motor processes [54]. In a clinical context, tDCS could be used to restore pathological brain functions and improve associated symptoms [55,56]. 

TDCS can generally be considered safe and well-tolerated. The safety of this technique was studied and tested by several researchers who concluded that tDCS, when used and monitored in accordance with international safety guidelines, was a safe and well-tolerated intervention [57]. Due to its relatively low costs and risks, it could be made available to a broad group of patients. Thus, tDCS has the potential to improve and enhance the quality of life by granting less limited access to a wider group of patients. Especially since costs are generally a key element limiting access to medicines, tDCS can substantially improve fairness in medical care.

In general, tDCS delivers small electrical currents to the cerebral cortex. The current flows between an active electrode and a reference electrode. While the scalp shunts some of this current, the majority enters the brain tissue (e.g., [58]), modulating cortical excitability [59]. Based on animal data [60] and seminal work on the human motor domain [61], a rather heuristic model for the mechanism of action was established. According to this somatic doctrine, the direction of the tDCS-induced effect depended on the current polarity. Anodal tDCS had an excitatory effect, while cathodal tDCS decreased cortical excitability in the region under the electrode [62]. These effects were mediated by depolarization of the resting membrane potential. Thus, anodal tDCS increased the neuronal firing rate due to a hyperpolarization of the resting membrane potential, while cathodal tDCS decreased the firing rate due to hypopolarization of the resting membrane potential. However, in contrast to studies examining tDCS effects on the primary motor cortex, the majority of tDCS studies challenged the somatic doctrine with conflicting [63,64] or opposing [65,66,67] anodal/cathodal effects. While these effects could be partially attributed to the nonlinear nature of the stimulation effects [59], neuroanatomy and, more specifically, the orientation of the somatodendritic axis within the stimulated cortical areas also seemed to be crucial [68]. Indeed, the somatic doctrine was based only on radially directed electric currents [69], but tDCS always generated significant tangential current flow due to cortical folding [70]. Thus, results from several tDCS studies underscored that findings of the underlying neural mechanisms obtained at the primary motor cortex could not simply be generalized to the broader cortical area (e.g., [71]). Interestingly, using a human neuronal in vitro model with a dopaminergic phenotype, a recent study showed that DCS exerts on-line and of-line effects on the expression, aggregation, and autophagic degradation of alpha-synuclein, indicating a potential neuroprotective role of tDCS [72]. 

## 4. Transcranial Direct Current Stimulation as a Therapeutic Option for Fatigue

The majority of the stimulation studies, designed to counteract the development of fatigue, applied anodal tDCS over the dorsolateral prefrontal cortex (DLPFC), as this area had proven to be most affected by fatigue [73,74,75,76,77,78,79].

In healthy participants, positive effects of anodal tDCS over the left DLPFC were consistently demonstrated. A single dose of anodal tDCS was able to reduce fatigue-related vigilance performance decrements over time [73,80], even more effectively than caffeine consumption was able to do [75,76]. Moreover, anodal tDCS over the left DLPFC could successfully counteract fatigability development and reduce the fatigability-related increase in occipital alpha power as well as the decline in sensory gating [77]. In this recent study, we demonstrated that a single session of prefrontal tDCS attenuated the fatigue-induced increase in occipital alpha power. We hypothesized that this effect might be related to a tDCS-induced increase in prefrontal theta power, as previously shown [79,80], supporting the proposed accentuated role of frontomedial theta power in compensatory control mechanisms to augment top-down control processes in a fatigued brain [48].

For MS-related fatigue, positive stimulation effects on subjective fatigue assessed with self-report scales were also reported after five consecutive days of anodal tDCS over the bilateral motor cortex or somatosensory cortex [81,82], over the left DLPFC [83], and bifrontal over the left and right DLPFC [8]. The observed tDCS-related improvement was greater in patients with a higher lesion load in the left frontal cortex [8]). Accordingly, long-term studies in which left frontal tDCS was applied consecutively for 4–6 weeks showed improvement in subjective fatigue that persisted up to 3 weeks thereafter [74,83]. Finally, also a single dose of tDCS over the left prefrontal cortex was an effective therapeutic option for treating fatigue-related deterioration in MS patients’ cognitive performance [84]. In this study, we investigated the effects of tDCS on fatigue development in patients with MS and demonstrated a positive effect of frontal tDCS. Anodal tDCS counteracted fatigue-associated performance decrements and improved patients’ ability to cope with sustained cognitive demands. The results suggested that tDCS-induced modulations of frontal activity may be an effective therapeutic option for treating fatigue-related deterioration of cognitive performance in patients with MS (see [74,85] for recent reviews).

In PD, applications of tDCS showed to be able to produce transient beneficial effects, both in the motor [86] as well in the non-motor domains, particularly on cognition [87].

Nowadays, however, reports of positive effects of tDCS on PDRF are only very sparse. In a first experiment, Forogh and colleagues [88] investigated the effect of multisession anodal tDCS over the left DLPFC on fatigue and daytime sleepiness in patients with PD. The authors applied a bilateral stimulation scheme with an anode over the left and a cathode over the right DLPFC and performed eight sessions of 20 min stimulation at a current of 0.06 mA/cm^2^ in 12 patients in an active treatment group and 11 patients in a placebo group. The data showed that anodal tDCS reduced fatigue immediately after treatment and also after a 3-month follow-up. As a further development of this approach, Dobbs and colleagues [89] proposed applying a remotely supervised tDCS protocol (RS-tDCS) to treat PDRF. The authors showed that a repeated application of anodal tDCS over the left DLPFC in a home-treatment context was well tolerated and positively affected subjective fatigue in patients with PD. Interestingly, the administration of repetitive transcranial magnetic stimulation (TMS) was found to improve motor and non-motor symptoms in patients with PD as well [90]. However, the majority of these studies assessed the effects of TMS on the excitability and plasticity of the motor cortex in patients with PD. Only sparse data also indicated supportive effects of TMS over the DLPFC on cognition [91] and depression [92]. 

In summary, PDRF is one of the most common non-motor symptoms occurring in the majority of patients and affecting a wide range of daily activities. PDRF results in a significant disability and markedly reduces the quality of life. The underlying pathophysiological mechanism in fatigue includes specific dysfunctions of the frontal cortex. However, despite its clinical importance, progress in developing an effective treatment for PDRF is still remarkably limited. Frontal anodal tDCS has proven to be effective for treating fatigue in both healthy participants and patients with neurological disorders such as multiple sclerosis. Moreover, anodal tDCS has been shown to raise hypofunctionality within stimulated cortical areas, including the DLPFC. Accordingly, the use of frontal anodal tDCS holds the promise of a potential therapeutic option for the treatment of PDRF. Further research is needed to determine the parameters of an optimal stimulation as well as to complement the purely subjective measures of fatigue with ones that provide an objective and valid assessment of fatigue and its potential reduction during treatment to make it useful in clinical settings. The concurrent use of neuroimaging methods such as EEG/MEG and fMRI in combination with tDCS is warranted and may be helpful in both target identification and outcome assessment for future tDCS trials for the treatment of PDRF. To conclude that frontal anodal tDCS can be an effective approach for the treatment of PDRF in a clinical setting, further data are needed that convincingly demonstrate (I) that a single session of anodal tDCS over the left DLPFC positively affects PDRF (transient effects), (II) that multisession tDCS can stabilize and/or enhance this effect, (III) that theses stimulation regimens lead to long-term effects of adequate duration, (IV) the specific conditions for a pronounced effect on the patient’s subjective as well as objective fatigue, and, finally, (IV) the specific parameters of a successful home-application.

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
