# Peer review of "Frontal Transcranial Direct Current Stimulation as a Potential Treatment of Parkinson’s Disease-Related Fatigue"

_brainsci, 2021, doi:10.3390/brainsci11040467_

Round 1

Reviewer 1 Report

The Author presents a comprehensive, well-written narrative review on the use of frontal tDCS as a potential treatment of PD-related fatigue. The topic is of great interest and innovative since fatigue is a disabling, not uncommon non-motor symptom in PD, and it has, in fact, no treatment.

The landscape of PD-related fatigue is complete and clear. The second part of the review reports on the fundamentals of tDCS and its rationale for its therapeutic use in PD and, specifically, in PD-related fatigue. At the end of the review, the Author reports two studies supporting the implementation of anodal tDCS applied in the DLFPC for PD-related fatigue.

Thus, the present article is more promotion for future research focusing on the treatment of PD-related fatigue by non-invasive brain stimulation.

To improve the interest of the study and make it more focused, I would suggest to reduce the section related to the general description of fatigue and add the following implementations:

-      Given the similar mechanism of action, I believe that summarizing the literature on the use of TMS for studying and treating PD-related fatigue would add significant value to the article;

-      Some studies on DBS treated patients also addressed the problem of fatigue. Maybe a brief overview on the deeper modulation of brain networks in PD patients suffering from fatigue could be added;

-      In the conclusions, I suggest adding more concrete suggestions to move forward the research on this topic by addressing potential issues and proposing examples of trials that could be performed in the next years.

Below, few minor comments:

Page 1, lines 35-36, lines 38-39, and lines 41-42. I believe that references are needed here

Page 2, lines 80-82. How is mood linked to fatigue here?

Reviewer 2 Report

Well written and concise paper, would benefit from a table

summarizing previous tDCS studies in the field of MS and PD

also discuss differences in settings, i.e. mA/fiedl intensity in the different studies

also a brief  discussion of pharmacologic treatments of fatigue in PD, as endorsed by the International Movement Disorders Society would be useful

Reviewer 3 Report

This short review by Tino Zaehle is very well written and describes a non-motor symptom often under-estimated, but physically and psychologically disabling, in patients with Parkinson's Disease and other movement disorders.

I have some concerns about the mechanisms of action of tDCS, as reported here, and about possible alternative targets than those extensively described in this paper.

1) The study did not cite some papers about the use of non-invasive brain stimulation (NIBS) techniques for the treatment of fatigue in Multiple Sclerosis (MS) and related diseases (e.g. Ferrucci et al., Neurorehabilitation 2015). That is of key importance to support the use of tDCS for fatigue in PD.

2) The section about the pathophysiology of fatigue is very well discussed. However, some targets are missing. For instance, the cerebellum is gaining a growing attention in the pathophysiology of both motor and non-motor symptoms in PD. The cerebellum istself expresses all the subtypes of dopaminergic receptors and is anatomically connected to the dorsolateral striatum thanks to a disynaptic pathway passing through the intralaminar nuclei of the thalamus (see Ferrucci et al., Cerebellum 2016). An increasing number of papers supports the use of the cerebellum as a target of NIBS (Bologna et al., Parkinsonism Relat Disord 2015; Workman et al., Brain Sci 2020).

3) More important, recent studies support the use of NIBS, particularly tDCS, as a disease-modifying therapy in neurodegenerative disorders. In particular, it has recently been shown that constant electrical fields reduces alpha-synuclein aggregation in in vitro models of PD, by promoting lysosomal degradation through the activation of both macro-autophagy and chaperone-mediated autophagy (Sala et al., Sci Rep 2021; see also Stefi et al., Pathophysiology 2019, and Osawa et al., Electrochemistry 2008).       

Round 2

Reviewer 1 Report

The Author has answered my previous questions and addressed my comments. I believe the study has improved and I have no further concerns.